# Towards an automatic Turing test: Learning to evaluate dialogue responses

**Ryan Lowe**$^{\heartsuit}$             **Michael Noseworthy**$^{\heartsuit *}$             **Iulian V. Serban**$^{\diamond *}$

**Nicolas Angelard-Gontier**$^{\heartsuit}$        **Yoshua Bengio**$^{\diamond \ddagger}$             **Joelle Pineau**$^{\heartsuit \ddagger}$

$^{\heartsuit}$ Reasoning and Learning Lab, School of Computer Science, McGill University
$^{\diamond}$ Montreal Institute for Learning Algorithms, Université de Montréal
$^{\ddagger}$ CIFAR Senior Fellow

## Abstract

Automatically evaluating the quality of dialogue responses for unstructured domains is a challenging problem. Unfortunately, existing automatic evaluation metrics are biased and correlate very poorly with human judgements of response quality (Liu et al., 2016). Yet having an accurate automatic evaluation procedure is crucial for dialogue research, as it allows rapid prototyping and testing of new models with fewer expensive human evaluations. In response to this challenge, we formulate automatic dialogue evaluation as a learning problem. We present an evaluation model (ADEM) that learns to predict human-like scores to input responses, using a new dataset of human response scores. We show that the ADEM model's predictions correlate significantly, and at level much higher than word-overlap metrics such as BLEU, with human judgements at both the utterance and system-level. We also show that ADEM can generalize to evaluating dialogue models unseen during training, an important step for automatic dialogue evaluation.

## 1 Introduction

Learning to communicate with humans is a crucial ability for intelligent agents. Among the primary forms of communication between humans is natural language dialogue. As such, building systems that can naturally and meaningfully converse with humans has been a central goal of artificial intelligence since the formulation of the Turing test (Turing, 1950). Research on one type of such systems, sometimes referred to as non-task-oriented dialogue systems, goes back to the mid-60s with Weizenbaum's famous program *ELIZA*: a rule-based system mimicking a Rogerian psychotherapist by persistently either rephrasing statements or asking questions (Weizenbaum, 1966). Recently, there has been a surge of interest in the research community towards building large-scale non-task-oriented dialogue systems using neural networks (Sordoni et al., 2015b; Shang et al., 2015; Vinyals & Le, 2015; Serban et al., 2016a; Li et al., 2015). These models are trained in an end-to-end manner to optimize a single objective, usually the likelihood of generating the responses from a fixed corpus. Such models have already had a substantial impact in industry, including Google's Smart Reply system (Kannan et al., 2016), and Microsoft's Xiaoice chatbot (Markoff & Mozur, 2015), which has over 20 million users. More recently, Amazon has announced the Alexa Prize Challenge: a research competition with the goal of developing a natural and engaging chatbot system (Farber, 2016).

One of the challenges when developing such systems is to have a good way of measuring progress, in this case the performance of the chatbot. The Turing test provides one solution to the evaluation of dialogue systems, but there are limitations with its original formulation. The test requires live human interactions, which is expensive and difficult to scale up. Furthermore, the test requires carefully designing the instructions to the human interlocutors, in order to balance their behaviour and expectations so that different systems may be ranked accurately by performance. Although unavoidable, these instructions introduce bias into the evaluation measure. The more common approach of having humans evaluate the quality of dialogue system responses, rather than distinguish them from human responses, induces similar drawbacks in terms of time, expense, and lack of

---

$^{*}$The second and third authors contributed equally.

scalability. In the case of chatbots designed for specific conversation domains, it may also be difficult to find sufficient human evaluators with appropriate background in the topic (e.g. Lowe et al. (2015)).

Despite advances in neural network-based models, evaluating the quality of dialogue responses automatically remains a challenging and under-studied problem in the non-task-oriented setting. The most widely used metric for evaluating such dialogue systems is BLEU (Papineni et al., 2002), a metric measuring word overlaps originally developed for machine translation. However, it has been shown that BLEU and other word-overlap metrics are biased and correlate poorly with human judgements of response quality (Liu et al., 2016). There are many obvious cases where these metrics fail, as they are often incapable of considering the semantic similarity between responses (see Figure 1). Despite this, many researchers still use BLEU to evaluate their dialogue models (Ritter et al., 2011; Sordoni et al., 2015b; Li et al., 2015; Galley et al., 2015; Li et al., 2016a), as there are few alternatives available that correlate with human judgements. While human evaluation should always be used to evaluate dialogue models, it is often too expensive and time-consuming to do this for every model specification (for example, for every combination of model hyperparameters). Therefore, having an accurate model that can evaluate dialogue response quality automatically — what could be considered an *automatic Turing test* — is critical in the quest for building human-like dialogue agents.

| Context of Conversation |
| --- |
| Speaker A: Hey, what do you want to do tonight? |
| Speaker B: Why don't we go see a movie? |
| **Model Response** |
| Nah, let's do something active. |
| **Reference Response** |
| Yeah, the film about Turing looks great! |

Figure 1: Example where word-overlap scores (e.g. BLEU) fail for dialogue evaluation; although the model response is completely reasonable, it has no words in common with the reference response, and thus would be given low scores by metrics such as BLEU.

To make progress towards this goal, we first collect a dataset of human scores to various dialogue responses, and we use this dataset to train an *automatic dialogue evaluation model*, which we call ADEM. The model is trained in a semi-supervised manner using a hierarchical recurrent neural network (RNN) to predict human scores. We show that ADEM scores correlate significantly, and at a level much higher than BLEU, with human judgement at both the utterance-level and system-level. Crucially, we also show that ADEM can generalize to evaluating new models, whose responses were unseen during training, without a drop in performance, making ADEM a strong first step towards effective automatic dialogue response evaluation.[1]

## 2 A DATASET FOR DIALOGUE RESPONSE EVALUATION

To train a model to predict human scores to dialogue responses, we first collect a dataset of human judgements (scores) of Twitter responses using the crowdsourcing platform Amazon Mechanical Turk (AMT).[2] The aim is to have accurate human scores for a variety of conversational responses — conditioned on dialogue contexts – which span the full range of response qualities. For example, the responses should include both relevant and irrelevant responses, both coherent and non-coherent responses and so on. To achieve this variety, we use candidate responses from several different models. Following Liu et al. (2016), we use the following 4 sources of candidate responses: (1) a response selected by a TF-IDF retrieval-based model, (2) a response selected by the Dual Encoder (DE) (Lowe et al., 2015), (3) a response generated using the hierarchical recurrent encoder-decoder (HRED) model (Serban et al., 2016a), and (4) human-generated responses. It should be noted that the human-generated candidate responses are *not* the

| | |
| --- | --- |
| # Examples | 4104 |
| # Contexts | 1026 |
| # Training examples | 2,872 |
| # Validation examples | 616 |
| # Test examples | 616 |
| $\kappa$ score (inter-annotator correlation) | 0.63 |

Table 1: Statistics of the dialogue response evaluation dataset. Each example is in the form *(context, model response, reference response, human score)*.

reference responses from a fixed corpus, but novel human responses that are different from the reference. In addition to increasing response variety, this is necessary because we want our evaluation model to learn to compare the reference responses to the candidate responses.

---

[1]We will provide open-source implementations of the model upon publication.

[2]All data collection was conducted in accordance with the policies of the host institutions' ethics board.

We conducted two rounds of AMT experiments. We first asked AMT workers to provide a reasonable continuation of a Twitter dialogue (i.e. generate the next response given the context of a conversation). Each survey contained 20 questions, including an attention check question. Workers were instructed to generate longer responses, in order to avoid simple one-word responses. In total, we obtained approximately 2,000 human responses.

Second, we filtered these human-generated responses for potentially offensive language, and combined them with approximately 1,000 responses from each of the above models into a single set of responses. We then asked AMT workers to rate the overall quality of each response on a scale of 1 (low quality) to 5 (high quality). Each user was asked to evaluate 4 responses from 50 different contexts. We included four additional attention-check questions and a set of five contexts was given to each participant for assessment of inter-annotator agreement. We removed all users who either failed an attention check question or achieved a $\kappa$ inter-annotator agreement score lower than 0.2 (Cohen, 1968). The remaining evaluators had a median $\kappa$ score of 0.63, indicating moderate agreement. This is consistent with results from Liu et al. (2016). Dataset statistics are provided in Table 1.

| Measurement | $\kappa$ score |
|---|---|
| Overall | 0.63 |
| Topicality | 0.57 |
| Informativeness | 0.31 |
| Background | 0.05 |

Table 2: Median $\kappa$ inter-annotator agreement scores for various questions asked in the survey.

In initial experiments, we also asked humans to provide scores for topicality, informativeness, and whether the context required background information to be understandable. Note that we did not ask for fluency scores, as 3/4 of the responses were produced by humans (including the retrieval models). We found that scores for informativeness and background had low inter-annotator agreement (Table 2), and scores for topicality were highly correlated with the overall score (Pearson correlation of 0.72). Results on these auxiliary questions varied depending on the wording of the question. Thus, we continued our experiments by only asking for the overall score. We provide more details concerning the data collection in the Appendix, as it may aid others in developing effective crowdsourcing experiments.

To train evaluation models on human judgements, it is crucial that we obtain scores of responses that lie near the distribution produced by state-of-the-art models. This is why we use the Twitter Corpus (Ritter et al., 2011), as such models are pre-trained and readily available. Further, the set of topics discussed is quite broad — as opposed to the very specific Ubuntu Dialogue Corpus — and therefore the model should generalize better to other domains involving chit-chat. Finally, since it does not require domain specific knowledge (e.g. technical knowledge), it should be easy for AMT workers to annotate.

## 3   TECHNICAL BACKGROUND

### 3.1   RECURRENT NEURAL NETWORKS

Recurrent neural networks (RNNs) are a type of neural network with time-delayed connections between the internal units. This leads to the formation of a *hidden state* $h_t$, which is updated for every input: $h_t = f(W_{hh}h_{t-1} + W_{ih}x_t)$, where $W_{hh}$ and $W_{ih}$ are parameter matrices, $f$ is a smooth non-linear activation function such as tanh, and $x_t$ is the input at time $t$. The hidden state allows for RNNs to better model sequential data, such as natural language.

In this paper, we consider RNNs augmented with long-short term memory (LSTM) units (Hochreiter & Schmidhuber, 1997). LSTMs add a set of gates to the RNN that allow it to learn how much to update the hidden state. LSTMs are one of the most well-established methods for dealing with the vanishing gradient problem in recurrent networks (Hochreiter, 1991; Bengio et al., 1994).

### 3.2   WORD-OVERLAP METRICS

One of the most popular approaches for automatically evaluating the quality of dialogue responses is by computing their *word overlap* with the reference response. In particular, the most popular metrics are the BLEU and METEOR scores used for machine translation, and the ROUGE score used for automatic summarization. While these metrics tend to correlate with human judgements in their target domains, they have recently been shown to highly biaqsed and correlate very poorly with

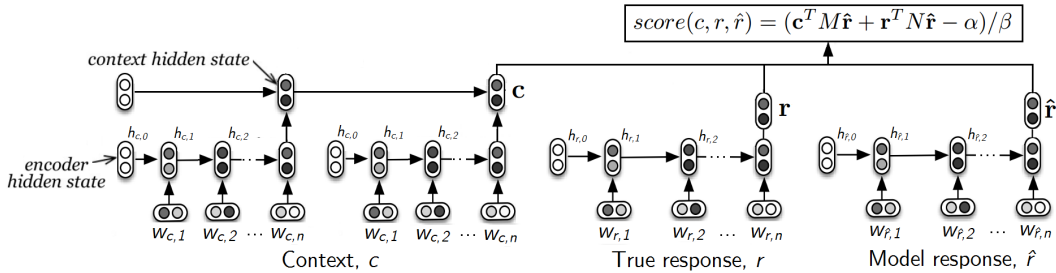

Figure 2: The ADEM model, which uses a hierarchical encoder to produce the context embedding **c**.

human judgements for dialogue response evaluation (Liu et al., 2016). We briefly describe BLEU here, and provide a more detailed summary of word-overlap metrics in the Appendix.

**BLEU**   BLEU (Papineni et al., 2002) analyzes the co-occurrences of n-grams in the ground truth and the proposed responses. It computes the n-gram precision for the whole dataset, which is then multiplied by a brevity penalty to penalize short translations. For BLEU-$N$, $N$ denotes the largest value of n-grams considered (usually $N = 4$).

**Drawbacks**   One of the major drawbacks of word-overlap metrics is their failure in capturing the semantic similarity between the model and reference responses when there are few or no common words. This problem is less critical for machine translation; since the set of reasonable translations of a given sentence or document is rather small, one can reasonably infer the quality of a translated sentence by only measuring the word-overlap between it and one (or a few) reference translations. However, in dialogue, the set of appropriate responses given a context is much larger (Artstein et al., 2009); in other words, there is a very high *response diversity* that is unlikely to be captured by word-overlap comparison to a single response.

Further, word-overlap scores are computed directly between the model and reference responses. As such, they do not consider the context of the conversation. While this may be a reasonable assumption in machine translation, it is not the case for dialogue; whether a model response is an adequate substitute for the reference response is clearly context-dependent. For example, the two responses in Figure 1 are equally appropriate given the context. However, if we simply change the context to: *"Have you heard of any good movies recently?"*, the model response is no longer relevant while the reference response remains valid.

## 4   AN AUTOMATIC DIALOGUE EVALUATION MODEL (ADEM)

To overcome the problems of evaluation with word-overlap metrics, we aim to construct a dialogue evaluation model that: (1) captures semantic similarity beyond word overlap statistics, and (2) exploits both the context of the conversation and the reference response to calculate its score for the model response. We call this evaluation model ADEM.

ADEM learns distributed representations of the context, model response, and reference response using a hierarchical RNN encoder. Given the dialogue context $c$, reference response $r$, and model response $\hat{r}$, ADEM first encodes each of them into vectors (**c**, $\hat{\mathbf{r}}$, and **r**, respectively) using the RNN encoder. Then, ADEM computes the score using a dot-product between the vector representations of $c$, $r$, and $\hat{r}$ in a linearly transformed space: :

$$score(c, r, \hat{r}) = (\mathbf{c}^T M \hat{\mathbf{r}} + \mathbf{r}^T N \hat{\mathbf{r}} - \alpha)/\beta \qquad (1)$$

where $M, N \in \mathbb{R}^n$ are learned matrices initialized to the identity, and $\alpha, \beta$ are scalar constants used to initialize the model's predictions in the range $[0, 5]$. The model is shown in Figure 2.

The matrices $M$ and $N$ can be interpreted as linear projections that map the model response $\hat{\mathbf{r}}$ into the space of contexts and reference responses, respectively. The model gives high scores to responses that have similar vector representations to the context and reference response after this projection. The model is end-to-end differentiable; all the parameters can be learned by backpropagation. In our

implementation, the parameters $\theta = \{M, N\}$ of the model are trained to minimize the squared error between the model predictions and the human score, with L1-regularization:

$$\mathcal{L} = \sum_{i=1:K} [score(c_i, r_i, \hat{r}_i) - human\_score_i]^2 + \gamma ||\theta||_1 \qquad (2)$$

where $\gamma$ is a scalar constant. The simplicity of our model leads to both accurate predictions and fast evaluation time (see Appendix), which is important to allow rapid prototyping of dialogue systems.

The hierarchical RNN encoder in our model consists of two layers of RNNs (El Hihi & Bengio, 1995; Sordoni et al., 2015a). The lower-level RNN, the *utterance-level encoder*, takes as input words from the dialogue, and produces a vector output at the end of each utterance. The *context-level encoder* takes the representation of each utterance as input and outputs a vector representation of the context. This hierarchical structure is useful for incorporating information from early utterances in the context (Serban et al., 2016a). Following previous work, we take the last hidden state of the context-level encoder as the vector representation of the input utterance or context.

An important point is that the ADEM procedure above *is not a dialogue retrieval model*. The fundamental difference between ADEM and a dialogue model is that ADEM has access to the reference response. Thus, ADEM can compare a model's response to a known good response, which is significantly easier than inferring response quality from solely the context.

**Pre-training with VHRED** We would like an evaluation model that can make accurate predictions from few labeled examples, since these examples are expensive to obtain. We therefore employ semi-supervised learning, and use a pre-training procedure to learn the parameters of the encoder. In particular, we train the encoder as part of a neural dialogue model; we attach a third *decoder RNN* that takes the output of the encoder as input, and train it to predict the next utterance of a dialogue conditioned on the context.

The dialogue model we employ for pre-training is the latent variable hierarchical recurrent encoder-decoder (VHRED) model (Serban et al., 2016b). The VHRED model is an extension of the original hierarchical recurrent encoder-decoder (HRED) model (Serban et al., 2016a) with a turn-level stochastic latent variable. The dialogue context is encoded into a vector using our hierarchical encoder, and the VHRED then samples a Gaussian variable that is used to condition the decoder (see Appendix for further details). After training VHRED, we use the last hidden state of the context-level encoder, when $c$, $r$, and $\hat{r}$ are fed as input, as the vector representations for $\mathbf{c}$, $\mathbf{r}$, and $\hat{\mathbf{r}}$, respectively. We use representations from the VHRED model as it produces more diverse and coherent responses compared to its HRED counterpart.

Maximizing the likelihood of generating the next utterance in a dialogue is not only a convenient way of training the encoder parameters; it is also an objective that is consistent with learning useful representations of the dialogue utterances. Two context vectors produced by the VHRED encoder are similar if the contexts induce a similar distribution over subsequent responses; this is consistent with the formulation of the evaluation model, which assigns high scores to responses that have similar vector representations to the context. VHRED is also closely related to the skip-thought-vector model (Kiros et al., 2015), which has been shown to learn useful representations of sentences for many tasks, including semantic relatedness and paraphrase detection. The skip-thought-vector model takes as input a single sentence and predicts the previous sentence and next sentence. On the other hand, VHRED takes as input several consecutive sentences and predicts the next sentence. This makes it particularly suitable for learning long-term context representations.

## 5 EXPERIMENTS

### 5.1 EXPERIMENTAL PROCEDURE

In order to reduce the effective vocabulary size, we use byte pair encoding (BPE) (Gage, 1994; Sennrich et al., 2015), which splits each word into sub-words or characters. We also use layer normalization (Ba et al., 2016) for the hierarchical encoder, which we found worked better at the task of dialogue generation than the related recurrent batch normalization (Ioffe & Szegedy, 2015; Cooijmans et al., 2016). To train the VHRED model, we employed several of the same techniques found in Serban et al. (2016b) and Bowman et al. (2016): we drop words in the decoder with a fixed

| | Full dataset | | Test set | |
|---|---|---|---|---|
| Metric | **Spearman** | **Pearson** | **Spearman** | **Pearson** |
| BLEU-1 | 0.026 (0.102) | 0.055 (<0.001) | 0.036 (0.413) | 0.074 (0.097) |
| BLEU-2 | 0.039 (0.013) | 0.081 (<0.001) | 0.051 (0.254) | 0.120 (<0.001) |
| BLEU-3 | 0.045 (0.004) | 0.043 (0.005) | 0.051 (0.248) | 0.073 (0.104) |
| BLEU-4 | 0.051 (0.001) | 0.025 (0.113) | 0.063 (0.156) | 0.073 (0.103) |
| ROUGE | 0.062 (<0.001) | 0.114 (<0.001) | 0.096 (0.031) | 0.147 (<0.001) |
| METEOR | 0.021 (0.189) | 0.022 (0.165) | 0.013 (0.745) | 0.021 (0.601) |
| T2V | 0.140 (<0.001) | 0.141 (<0.001) | 0.140 (<0.001) | 0.141 (<0.001) |
| VHRED | -0.035 (0.062) | -0.030 (0.106) | -0.091 (0.023) | -0.010 (0.805) |
| | **Validation set** | | **Test set** | |
| C-ADEM | 0.272 (<0.001) | 0.238 (<0.001) | 0.293 (<0.001) | 0.303 (<0.001) |
| R-ADEM | 0.428 (<0.001) | 0.383 (<0.001) | 0.409 (<0.001) | 0.392 (<0.001) |
| ADEM (T2V) | 0.395 (<0.001) | **0.392** (<0.001) | 0.408 (<0.001) | **0.411** (<0.001) |
| ADEM | **0.436** (<0.001) | **0.389** (<0.001) | **0.414** (<0.001) | 0.395 (<0.001) |

Table 3: Correlation between metrics and human judgements, with p-values shown in brackets. 'ADEM (T2V)' indicates ADEM with tweet2vec embeddings (Dhingra et al., 2016), and 'VHRED' indicates the dot product of VHRED embeddings (i.e. ADEM at initialization). C- and R-ADEM represent the ADEM model trained to only compare the model response to the context or reference response, respectively.

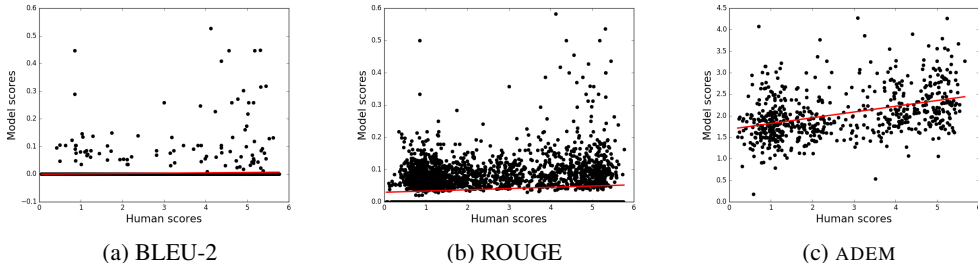

(a) BLEU-2                (b) ROUGE                (c) ADEM

Figure 3: Scatter plot showing model against human scores, for BLEU-2 and ROUGE on the full dataset, and ADEM on the test set. We add Gaussian noise drawn from $\mathcal{N}(0, 0.3)$ to the integer human scores to better visualize the density of points, at the expense of appearing less correlated.

rate of 25%, and we anneal the KL-divergence term linearly from 0 to 1 over the first 60,000 batches. We use Adam as our optimizer (Kingma & Ba, 2014).

For training VHRED, we use a context embedding size of 2000. However, we found the ADEM model learned more effectively when this embedding size was reduced. Thus, after training VHRED, we use principal component analysis (PCA) (Pearson, 1901) to reduce the dimensionality of the context, model response, and reference response embeddings to $n$. While our results are robust to $n$, we found experimentally that $n = 7$ provided slightly improved performance. We provide other hyperparameter values in the Appendix.

When evaluating our models, we conduct early stopping on a separate validation set to obtain the best parameter setting. For the evaluation dataset, we split the train/ validation/ test sets such that there is no context overlap (i.e. the contexts in the test set are unseen during training).

## 5.2 RESULTS

**Utterance-level correlations**   We first present new utterance-level correlation results[3] for existing word-overlap metrics, in addition to results with embedding baselines and ADEM, in Table 3. The

---

[3]We present both the Spearman correlation (computed on ranks, depicts monotonic relationships) and Pearson correlation (computed on true values, depicts linear relationships) scores.

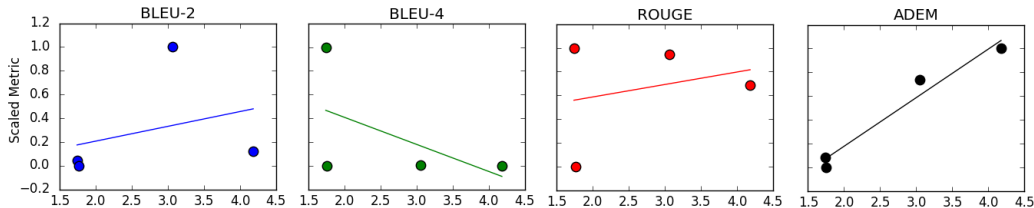

Figure 4: Scatterplots depicting the system-level correlation results for BLEU-2, BLEU-4, ROUGE, and ADEM on the test set. Each point represents the average scores for the responses from a dialogue model (TFIDF, DE, HRED, human). Human scores are shown on the horizontal axis, with normalized metric scores on the vertical axis. The ideal metric has a perfectly linear relationship.

baseline metrics are evaluated on the entire dataset of 4,104 responses.[4] We measure the correlation for ADEM on the validation and test sets (616 responses each).

We also conduct an additional analysis of the response data from Liu et al. (2016), where the pre-processing is standardized by removing '<first_speaker>' tokens at the beginning of each utterance. The results are detailed in Table 10 of Appendix D. We can observe from both this data, and the new data in Table 3, that the correlations for the word-overlap metrics are even lower than estimated in previous studies (Liu et al., 2016; Galley et al., 2015). In particular, this is the case for BLEU-4, which has frequently been used for dialogue response evaluation (Ritter et al., 2011; Sordoni et al., 2015b; Li et al., 2015; Galley et al., 2015; Li et al., 2016a).

We can see from Table 3 that ADEM correlates far better with human judgement than the word-overlap baselines. This is further illustrated by the scatterplots in Figure 3. We also compare with ADEM using tweet2vec embeddings for $\mathbf{c}$, $\mathbf{r}$, and $\hat{\mathbf{r}}$, which are computed at the character-level with a bidirectional GRU (Dhingra et al., 2016), and obtain comparable but slightly inferior performance compared to using VHRED embeddings.

**System-level correlations** We show the system-level correlations for various metrics in Table 4, and present it visually in Figure 4. Each point in the scatterplots represents a dialogue model; humans give low scores to TFIDF and DE responses, higher scores to HRED and the highest scores to other human responses. It is clear that existing word-overlap metrics are incapable of capturing this relationship for even 4 models. This renders them *completely deficient* for dialogue evaluation. However, ADEM produces the exact same model ranking as humans, achieving a significant Pearson correlation of 0.98.[5] Thus, ADEM correlates well with humans both at the response and system level.

| Metric | Pearson |
|--------|---------|
| BLEU-1 | -0.079 (0.921) |
| BLEU-2 | 0.308 (0.692) |
| BLEU-3 | -0.537 (0.463) |
| BLEU-4 | -0.536 (0.464) |
| ROUGE | 0.268 (0.732) |
| ADEM | **0.981** (0.019) |

Table 4: System-level correlation, with the p-value in brackets.

**Generalization to previously unseen models** When ADEM is used in practice, it will take as input responses from a new model that it has not seen during training. Thus, it is crucial that ADEM correlates with human judgements for new models. We test ADEM's generalization ability by performing a leave-one-out evaluation. For each dialogue model that was the source of response data for training ADEM (TF-IDF, Dual Encoder, HRED, humans), we conduct an experiment where we train on all model responses *except* those from the chosen model, and test *only* on the model that was unseen during training.

The results are given in Table 5. Overall, we observe that the ADEM model is very robust, and is capable of generalizing to new models in all cases. When testing the correlation on the entire test set, the model achieves comparable correlations to the ADEM model that was trained on 25%

---

[4]Note that our word-overlap correlation results in Table 3 are also lower than those presented in Galley et al. (2015). This is because Galley et al. measure corpus-level correlation, i.e. correlation averaged across different subsets (of size 100) of the data, and pre-filter for high-quality reference responses.

[5]For comparison, BLEU achieves a system-level correlation of 0.99 on 5 models in the translation domain (Papineni et al., 2002).

| | Test on full dataset | | Test on removed model responses | |
|---|---|---|---|---|
| Data Removed | Spearman | Pearson | Spearman | Pearson |
| TF-IDF | 0.4097 (<0.001) | 0.3975 (<0.001) | 0.3931 (<0.001) | 0.3645 (<0.001) |
| Dual Encoder | 0.4000 (<0.001) | 0.3907 (<0.001) | 0.4256 (<0.001) | 0.4098 (<0.001) |
| HRED | 0.4128 (<0.001) | 0.3961 (<0.001) | 0.3998 (<0.001) | 0.3956 (<0.001) |
| Human | 0.4052 (<0.001) | 0.3910 (<0.001) | 0.4472 (<0.001) | 0.4230 (<0.001) |
| **Average** | **0.4069** (<0.001) | **0.3938** (<0.001) | **0.4164** (<0.001) | **0.3982** (<0.001) |
| 25% at random | 0.4077 (<0.001) | 0.3932 (<0.001) | — | — |

Table 5: Correlation for ADEM when various model responses are removed from the training set. The left two columns show performance on the entire test set, and the right two columns show performance on responses only from the dialogue model not seen during training. The last row (25% at random) corresponds to the ADEM model trained on all model responses, but with the same amount of training data as the model above (i.e. 25% less data than the full training set).

| Context | Reference response | Model responses | Human score | ADEM score |
|---|---|---|---|---|
| photo to see my television debut go to - some. some on <url> - hehe <url> → it really was you? i thought ppl were recognizing someone who looked like you! were the oysters worth the wait? | yeah it was me . haha i 'd kinda forgotten about it it was filmed a while ago | 1) i'm not sure. i just don't know what to do with it. | 3 | 1.602 |
| | | 2) you heard the horsepower productions remix of lee scratch perry's 'exercising' off his 'mighty upsetter' album? | 1 | 1.513 |
| | | 3) you wont chug a fuzzy peach navel | 1 | 1.744 |
| | | 4) they were! | 5 | 3.274 |
| just beat call of duty!! → want a cookie? → yes!! → come get it | im in kenmore at the moment | 1) i'm gonna get a new phone some moro | 1 | 1.848 |
| | | 2) no way man. | 5 | 4.265 |
| | | 3) wow i just got a free pizza coupon! get yours before theres no more! <url> | 1 | 0.921 |
| | | 4) i'm going to go to the mall. | 1 | 2.634 |
| am i out of twitter jail yet? testing → yeah. i posted bail → thanks. i am a right chatter tweetbox on sundays. same happened last sunday lol | any news on meeting our user ? i go to the us on friday and i don 't want to miss anything arranged | 1) i'm not sure if i'm going to be able to get it. | 3 | 1.912 |
| | | 2) good to see another mac user in the leadership ranks | 4 | 1.417 |
| | | 3) awww poor baby hope u get to feeling better soon. maybe some many work days at piedmont | 2 | 1.123 |
| | | 4) did you tweet too much? | 5 | 2.539 |

Table 7: Examples of scores given by the ADEM model.

less data selected at random. This is particularly surprising for the HRED model; in this case, ADEM was trained only on responses that were written by humans (from retrieval models or human-generated), but is able to generalize to responses produced by a generative neural network model. This demonstrates ADEM's ability to accurately score new neural network-based dialogue models.

**Qualitative Analysis**    To illustrate some strengths and weaknesses of ADEM, we show human and ADEM scores for each of the responses to various contexts in Table 7. There are several instances where ADEM predicts accurately: in particular, ADEM is often very good at assigning low scores to poor responses. This seen in the first two contexts, where most of the responses given a score of 1 from humans are given scores less than 2 by ADEM. The single exception in response (4) for the second context seems somewhat appropriate and should perhaps have been scored higher by the human evaluator. There are also several instances where the model assigns high scores to suitable responses, as in the first two contexts.

One drawback we observed is that ADEM tends to be too conservative when predicting response scores. This is the

| Metric scores | # Examples |
|---|---|
| Human ≥ 4 | 237 out of 616 |
| **and** (\|BLEU-2\| <2, \|ROUGE\| <2) | 146 out of 237 |
| **and** \|ADEM\| > 4 | 60 out of 146 |
| **and** \|ADEM\| < 2 | 42 out of 237 |
| **and** (\|BLEU-2\| >4, **or** \|ROUGE\| >4) | 14 out of 42 |

Table 6: In 60/146 cases, ADEM scores good responses (human score > 4) highly when word-overlap metrics fail. The bars around |metric| indicate that the metric scores have been normalized.

case in the third context, where the model assigns low scores to most of the responses that a human rated highly (although response (2) is arguably not relevant to the context). This behaviour is likely due to the squared error loss used to train ADEM; since the model receives a large penalty for incorrectly predicting an extreme value, it learns to predict scores closer to the average human score.

| Context | Reference response | Model response | Human score | \|BLEU-2\| score | \|ROUGE\| score | \|ADEM\| score |
|---|---|---|---|---|---|---|
| i'd recommend \<url\> - or build buy an htpc and put \<url\> on it. → you're the some nd person this week that's recommended roku to me. | an htpc with xmbc is what i run . but i 've decked out my setup . i 've got \<number\> tb of data on my home server | because it's brilliant | 5 | 1.0 | 1.0 | 4.726 |
| imma be an auntie this weekend. i guess i have to go albany. herewego → u supposed to been here → i come off nd on. → never tell me smh | lol you sometiming | haha, anyway, how're you? | 5 | 1.0 | 1.0 | 4.201 |
| my son thinks she is plain. and the girl that plays her sister. seekhelp4him? → send him this. he'll thank you. \<url\> | you are too kind for words . | i will do | 5 | 1.0 | 1.0 | 5.0 |

Table 9: Examples where both human and ADEM score the model response highly, while BLEU-2 and ROUGE do not. These examples are drawn randomly (i.e. no cherry-picking) from the examples where ADEM outperforms BLEU-2 and ROUGE (as defined in the text). ADEM is able to correctly assign high scores to short responses that have no word-overlap with the reference response. The bars around |metric| indicate that the metric scores have been normalized.

**Correlation with response length**    One implicit assumption in the ADEM model is that the human evaluations of model responses is absolutely correct, including the biases that humans exhibit when evaluating dialogues. For example, it has been shown that humans have a tendency to give a higher rating to shorter responses than to longer responses (Serban et al., 2016b), as shorter responses are often more generic and thus are more likely to be suitable to the context. This affects dialogue response evaluation: we calculated the test set correlation between response length and the human score, and obtained a significant Pearson correlation of 0.27, and a Spearman correlation of 0.32. If the assumption that human evaluators are absolutely correct is not accurate, it may be desirable to remove human biases in an automatic evaluation model to improve the model's generalization capabilities. This is an important direction for future work.

**Improvement over word-overlap metrics**    Next, we analyze more precisely how ADEM outperforms traditional word-overlap metrics such as BLEU-2 and ROUGE. We first normalize the metric scores to have the same mean and variance as human scores, clipping the resulting scores to the range $[1, 5]$ (we assign raw scores of 0 a normalized score of 1). *We indicate normalization with vertical bars around the metric.* We then select all of the good responses that were given low scores by word-overlap metrics (i.e. responses which humans scored as 4 or higher, and which |BLEU-2| and |ROUGE| scored as 2 or lower). The results are summarized in Table 6: of the 237 responses that humans scored 4 or higher, most of them (147/237) were ranked very poorly by both BLEU-2 and ROUGE. This quantitatively demonstrates what we argued qualitatively in Figure 1; a major failure of word-overlap metrics is the inability to consider reasonable responses that have no word-overlap with the reference response. We can also see that, in almost half (60/147) of the cases where both BLEU-2 and ROUGE fail, |ADEM| is able to correctly assign a score greater than 4. For comparison, there are only 42 responses where humans give a score of 4 and |ADEM| gives a score less than 2, and only 14 of these are assigned a score greater than 4 by either |BLEU-2| or |ROUGE|.

To provide further insight, we give specific examples of responses that are scored highly ($> 4$) by both humans and |ADEM|, and poorly ($< 2$) by both |BLEU-2| and |ROUGE| in Table 9. We draw 3 responses randomly (i.e. no cherry-picking) from the 60 test set responses that meet this criteria. We can observe that ADEM is able to recognize short responses that are appropriate to the context, without word-overlap with the reference response. This is even the case when the model and reference responses have very little semantic similarity, as in the first and third examples in Table 9.

Finally, we show the behaviour of ADEM when there is a discrepancy between the lengths of the reference and model responses. In (Liu et al., 2016), the authors show that word-overlap metrics such as BLEU-1, BLEU-2, and METEOR exhibit a bias in this scenario: they tend to assign higher scores to responses that are

| | **Mean score** | | |
|---|---|---|---|
| | $\Delta w \leq 6$ (n=312) | $\Delta w > 6$ (n=304) | **p-value** |
| ROUGE | 0.042 | 0.031 | $< 0.01$ |
| BLEU-2 | 0.0022 | 0.0007 | 0.23 |
| ADEM | 2.072 | 2.015 | 0.23 |
| Human | 2.671 | 2.698 | 0.83 |

Table 8: Effect of differences in response length on the score, $\Delta w$ = absolute difference in #words between the reference response and proposed response. BLEU-1, BLEU-2, and METEOR have previously been shown to exhibit bias towards similar-length responses (Liu et al., 2016).

closer in length to the reference response.[6] However, humans do not exhibit this bias; in other words, the quality of a response as judged by a human is roughly independent of its length. In Table 8, we show that ADEM also does not exhibit this bias towards similar-length responses.

## 6    RELATED WORK

Related to our approach is the literature on novel methods for the evaluation of machine translation systems, especially through the WMT evaluation task (Callison-Burch et al., 2011; Machácek & Bojar, 2014; Stanojevic et al., 2015). In particular, Gupta et al. (2015) have recently proposed to evaluate machine translation systems using Tree-LSTMs. Their approach differs from ours as, in the dialogue domain, we must additionally condition our score on the context of the conversation, which is not necessary in translation.

Several recent approaches use hand-crafted reward features to train dialogue models using reinforcement learning (RL). For example, Li et al. (2016b) use features related to ease of answering and information flow, and Yu et al. (2016) use metrics related to turn-level appropriateness and conversational depth. These metrics are based on hand-crafted features, which only capture a small set of relevant aspects; this inevitably leads to sub-optimal performance, and it is unclear whether such objectives are preferable over retrieval-based cross-entropy or word-level maximum log-likelihood objectives. Furthermore, many of these metrics are computed at the conversation-level, and are not available for evaluating single dialogue responses. The metrics that can be computed at the response-level could be incorporated into our framework, for example by adding a term to equation 1 consisting of a dot product between these features and a vector of learned parameters.

There has been significant work on evaluation methods for task-oriented dialogue systems, which attempt to solve a user's task such as finding a restaurant. These methods include the PARADISE framework (Walker et al., 1997) and MeMo (Möller et al., 2006), which consider a task completion signal. Our models do not attempt to model task completion, and thus fall outside this domain.

## 7    DISCUSSION

We use the Twitter Corpus to train our models as it contains a broad range of non-task-oriented conversations and has has been used to train many state-of-the-art models. However, our model could easily be extended to other general-purpose datasets, such as Reddit, once similar pre-trained models become publicly available. Such models are necessary even for creating a test set in a new domain, which will help us determine if ADEM generalizes to related dialogue domains. We leave investigating the domain transfer ability of ADEM for future work.

The evaluation model proposed in this paper favours dialogue models that generate responses that are rated as highly appropriate by humans. It is likely that this property does not fully capture the desired end-goal of chatbot systems. For example, one issue with building models to approximate human judgements of response quality is the problem of generic responses. Since humans often provide high scores to generic responses due to their appropriateness for many given contexts, a model trained to predict these scores will exhibit the same behaviour. An important direction for future work is modifying ADEM such that it is not subject to this bias. This could be done, for example, by censoring ADEM's representations (Edwards & Storkey, 2016) such that they do not contain any information about length. Alternatively, one could build a second evaluation model that assigns a score based on how easy it is to distinguish the dialogue model responses from human responses. In this case, a model that generates generic responses will easily be distinguishable and obtain a low score.

An important direction of future research is building models that can evaluate the capability of a dialogue system to have an engaging and meaningful interaction with a human. Compared to evaluating a single response, this evaluation is arguably closer to the end-goal of chatbots. However, such an evaluation is extremely challenging to do in a completely automatic way. We view the evaluation procedure presented in this paper as an important step towards this goal; current dialogue systems are incapable of generating responses that are rated as highly appropriate by humans, and we believe our evaluation model will be useful for measuring and facilitating progress in this direction.

---

[6]Note that, for our dataset, BLEU-2 almost exclusively assigns scores near 0 for both $\Delta w \leq 6$ and $\Delta w > 6$, resulting in a p-value $>0.05$.

ACKNOWLEDGEMENTS

We'd like to thank Casper Liu for his help with the correlation code, Laurent Charlin for helpful discussions on the data collection, Jason Weston for suggesting improvements in the experiments, and Jean Harb and Emmanuel Bengio for their debugging skills. We gratefully acknowledge support from the Samsung Institute of Advanced Technology, the National Science and Engineering Research Council, and Calcul Quebec. We'd also like to thank the developers of Theano (Team et al., 2016).

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

APPENDIX A: FURTHER NOTES ON CROWDSOURCING DATA COLLECTION

Before conducting the primary crowdsourcing experiments to collect the dataset in this paper, we ran a series of preliminary experiments to see how AMT workers responded to different questions. Unlike the primary study, where we asked a small number of overlapping questions to determine the $\kappa$ score and filtered users based on the results, we conducted a study where all responses (40 in total from 10 contexts) were overlapping. We did this for 18 users in two trials, resulting in 153 pair-wise correlation scores per trial.

In the first trial, we asked the following questions to the users, for each response:

1. How appropriate is the response overall? (overall, scale of 1-5)
2. How on-topic is the response? (topicality, scale of 1-5)
3. How specific is the response to some context? (specificity, scale of 1-5)
4. How much background information is required to understand the context? (background, scale of 1-5)

Note that we do not ask for fluency, as the 3/4 responses for each context were written by a human (including retrieval models). We also provided the AMT workers with examples that have high topicality and low specificity, and examples with high specificity and low topicality. The background question was only asked once for each context.

We observed that both the overall scores and topicality had fairly high inter-annotator agreement (as shown in Table 2), but were strongly correlated with each other (i.e. participants would often put the same scores for topicality and overall score). Conversely, specificity ($\kappa = 0.12$) and background ($\kappa = 0.05$) had very low inter-annotator agreements.

To better visualize the data, we produce scatterplots showing the distribution of scores for different responses, for each of the four questions in our survey (Figure 5). We can see that the overall and topicality scores are clustered for each question, indicating high agreement. However, these clusters are most often in the same positions for each response, which indicates that they are highly correlated with each other. Specificity and background information, on the other hand, show far fewer clusters, indicating lower inter-annotator agreement. We conjectured that this was partially because the terms 'specificity' and 'background information', along with our descriptions of them, had a high cognitive load, and were difficult to understand in the context of our survey.

To test this hypothesis, we conducted a new survey where we tried to ask the questions for specificity and background in a more intuitive manner. We also changed the formulation of the background question to be a binary 0-1 decision of whether users understood the context. We asked the following questions:

1. How appropriate is the response overall? (overall, scale of 1-5)
2. How on-topic is the response? (topicality, scale of 1-5)
3. How common is the response? (informativeness, scale of 1-5)
4. Does the context make sense? (context, scale of 0-1)

We also clarified our description for the third question, including providing more intuitive examples. Interestingly, the inter-annotator agreement on informativeness $\kappa = 0.31$ was much higher than that for specificity in the original survey. Thus, the formulation of questions in a crowdsourcing survey has a large impact on inter-annotator agreement. For the context, we found that users either agreed highly ($\kappa > 0.9$ for 45 participants), or not at all ($\kappa < 0.1$ for 113 participants).

We also experimented with asking the overall score on a separate page, before asking questions 2-4, and found that this increased the $\kappa$ agreement slightly. Similarly, excluding all scores where participants indicated they did not understand the context improved inter-annotator agreement slightly.

Due to these observations, we decided to only ask users for their overall quality score for each response, as it is unclear how much additional information is provided by the other questions in the context of dialogue. We hope this information is useful for future crowdsourcing experiments in the dialogue domain.

**Overall**

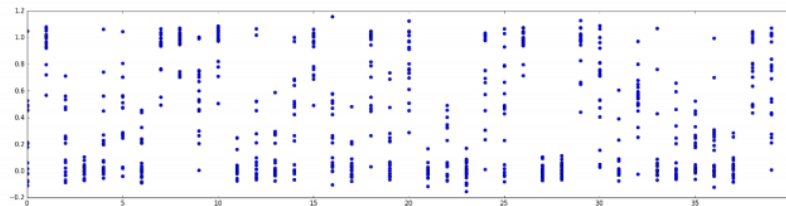

**Topicality**

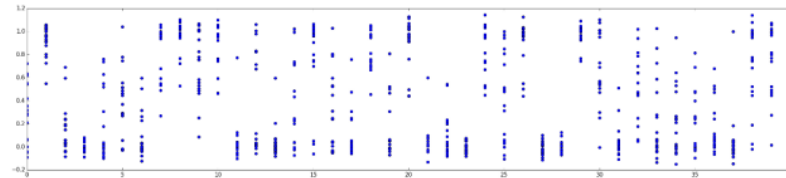

**Specificity**

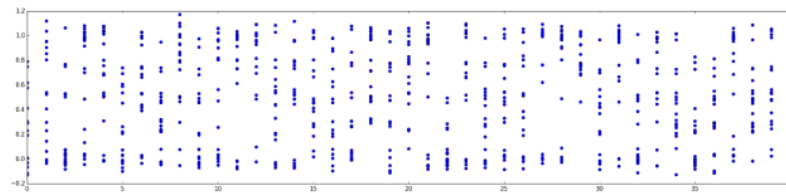

**Background**

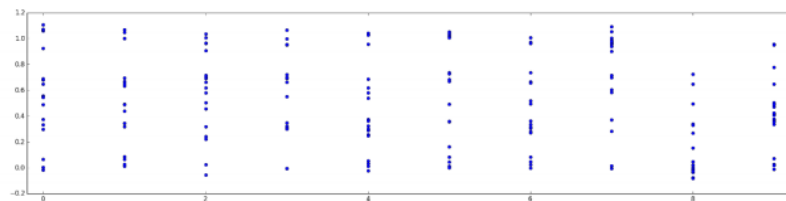

Figure 5: Scatter plots showing the distribution of scores (vertical axis) for different responses (horizontal axis), for each of the four questions in our survey. It can be seen that the overall and topicality scores are clustered for each question, indicating high agreement, while this is not the case for specificity or background information. Note that all scores are normalized based on a per-user basis, based on the average score given by each user.

APPENDIX B: METRIC DESCRIPTION

**BLEU**  BLEU (Papineni et al., 2002) analyzes the co-occurrences of n-grams in the ground truth and the proposed responses. It first computes an n-gram precision for the whole dataset:

$$P_n(r, \hat{r}) = \frac{\sum_k \min(h(k, r), h(k, \hat{r}_i))}{\sum_k h(k, r_i)}$$

where $k$ indexes all possible n-grams of length $n$ and $h(k, r)$ is the number of n-grams $k$ in $r$. Note that the min in this equation is calculating the number of co-occurrences of n-gram $k$ between the ground truth response $r$ and the proposed response $\hat{r}$, as it computes the fewest appearances of $k$ in either response. To avoid the drawbacks of using a precision score, namely that it favours shorter (candidate) sentences, the authors introduce a brevity penalty. BLEU-N, where $N$ is the maximum length of n-grams considered, is defined as:

$$\text{BLEU-N} := b(r, \hat{r}) \exp(\sum_{n=1}^{N} \beta_n \log P_n(r, \hat{r}))$$

$\beta_n$ is a weighting that is usually uniform, and $b(\cdot)$ is the brevity penalty. The most commonly used version of BLEU assigns $N = 4$. Modern versions of BLEU also use sentence-level smoothing, as the geometric mean often results in scores of 0 if there is no 4-gram overlap (Chen & Cherry, 2014). Note that BLEU is usually calculated at the corpus-level, and was originally designed for use with multiple reference sentences.

**METEOR**  The METEOR metric (Banerjee & Lavie, 2005) was introduced to address several weaknesses in BLEU. It creates an explicit alignment between the candidate and target responses. The alignment is based on exact token matching, followed by WordNet synonyms, stemmed tokens, and then paraphrases. Given a set of alignments, the METEOR score is the harmonic mean of precision and recall between the proposed and ground truth sentence.

Given a set of alignments $m$, the METEOR score is the harmonic mean of precision $P_m$ and recall $R_m$ between the candidate and target sentence.

$$Pen = \gamma(\frac{ch}{m})^\theta \tag{3}$$

$$F_{mean} = \frac{P_m R_m}{\alpha P_m + (1 - \alpha) R_m} \tag{4}$$

$$P_m = \frac{|m|}{\sum_k h_k(c_i)} \tag{5}$$

$$R_m = \frac{|m|}{\sum_k h_k(s_{ij})} \tag{6}$$

$$METEOR = (1 - Pen) F_{mean} \tag{7}$$

The penalty term $Pen$ is based on the 'chunkiness' of the resolved matches. We use the default values for the hyperparameters $\alpha, \gamma$, and $\theta$.

**ROUGE**  ROUGE (Lin, 2004) is a set of evaluation metrics used for automatic summarization. We consider ROUGE-L, which is a F-measure based on the Longest Common Subsequence (LCS) between a candidate and target sentence. The LCS is a set of words which occur in two sentences in the same order; however, unlike n-grams the words do not have to be contiguous, i.e. there can be other words in between the words of the LCS. ROUGE-L is computed using an F-measure between the reference response and the proposed response.

$$R = \max_j \frac{l(c_i, s_{ij})}{|s_{ij}|} \tag{8}$$

$$P = \max_j fracl(c_i, s_{ij})|c_{ij}| \tag{9}$$

$$ROUGE_L(c_i, S_i) = \frac{(1 + \beta^2)RP}{R + \beta^2 P} \tag{10}$$

where $l(c_i, s_{ij})$ is the length of the LCS between the sentences. $\beta$ is usually set to favour recall ($\beta = 1.2$).

## APPENDIX C: LATENT VARIABLE HIERARCHICAL RECURRENT ENCODER-DECODER (VHRED)

The VHRED model is an extension of the original hierarchical recurrent encoder-decoder (HRED) model (Serban et al., 2016a) with an additional component: a high-dimensional stochastic latent variable at every dialogue turn. The dialogue context is encoded into a vector representation using the *utterance-level* and *context-level* RNNs from our encoder. Conditioned on the summary vector at each dialogue turn, VHRED samples a multivariate Gaussian variable that is provided, along with the context summary vector, as input to the *decoder* RNN, which in turn generates the response word-by-word. We use representations from the VHRED model as it produces more diverse and coherent responses compared to its HRED counterpart.

The VHRED model is trained to maximize a lower-bound on the log-likelihood of generating the next response:

$$\mathcal{L} = \log P_{\hat{\theta}}(\mathbf{w}_1, \ldots, \mathbf{w}_N)$$
$$\geq \sum_{n=1}^{N} -\text{KL}\left[Q_\psi(\mathbf{z}_n \mid \mathbf{w}_1, \ldots, \mathbf{w}_n) || P_{\hat{\theta}}(\mathbf{z}_n \mid \mathbf{w}_{<n})\right] + \mathbb{E}_{Q_\psi(\mathbf{z}_n|\mathbf{w}_1,\ldots,\mathbf{w}_n)}\left[\log P_{\hat{\theta}}(\mathbf{w}_n \mid \mathbf{z}_n, \mathbf{w}_{<n})\right],$$

where $\text{KL}[Q||P]$ is the Kullback-Leibler (KL) divergence between distributions $Q$ and $P$. The distribution $Q_\psi(\mathbf{z}_n \mid \mathbf{w}_1, \ldots, \mathbf{w}_N) = \mathcal{N}(\boldsymbol{\mu}_{\text{posterior}}(\mathbf{w}_1, \ldots, \mathbf{w}_n), \Sigma_{\text{posterior}}(\mathbf{w}_1, \ldots, \mathbf{w}_n))$ is the approximate posterior distribution (or *recognition model*) which approximates the intractable true posterior distribution $P_\psi(\mathbf{z}_n \mid \mathbf{w}_1, \ldots, \mathbf{w}_N)$. The posterior mean $\boldsymbol{\mu}_{\text{posterior}}$ and covariance $\Sigma_{\text{posterior}}$ (as well as that of the prior) are computed using a feed-forward neural network, which takes as input the concatenation of the vector representations of the past utterances and that of the current utterance.

The multivariate Gaussian latent variable in the VHRED model allows modelling ambiguity and uncertainty in the dialogue through the latent variable distribution parameters (mean and variance). This provides a useful inductive bias, which helps VHRED encode the dialogue context into a real-valued embedding space even when the dialogue context is ambiguous or uncertain, and it helps VHRED generate more diverse responses.

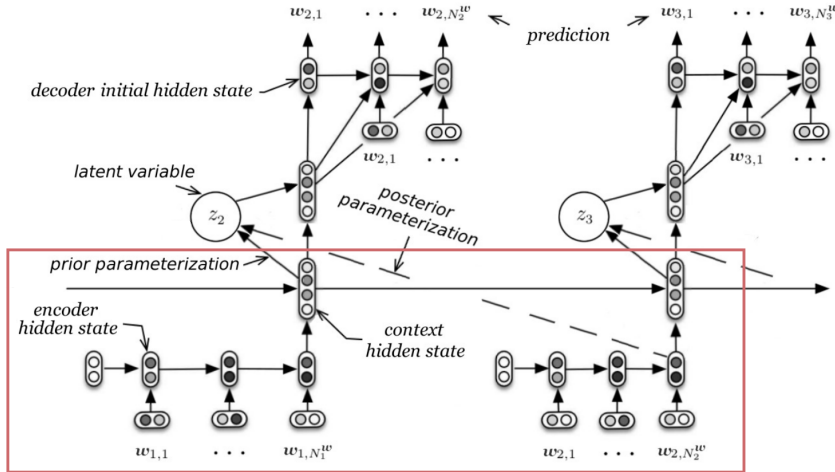

Figure 6: The VHRED model used for pre-training. The hierarchical structure of the RNN encoder is shown in the red box around the bottom half of the figure. After training using the VHRED procedure, the last hidden state of the context-level encoder is used as a vector representation of the input text.

## APPENDIX D: EXPERIMENTS & RESULTS

### HYPERPARAMETERS

When evaluating our model, we conduct early stopping on an external validation set to obtain the best parameter setting. We similarly choose our hyperparameters (PCA dimension $n$, L1 regularization penalty $\gamma$, learning rate $a$, and batch size $b$) based on validation set results. Our best ADEM model used $\gamma = 0.02$, $a = 0.01$, and $b = 16$. For ADEM with tweet2vec embeddings, we did a similar hyperparameter searched, and used $n = 150$, $\gamma = 0.01$, $a = 0.01$, and $b = 16$.

### ADDITIONAL RESULTS

**New results on Liu et al. (2016) data** In order to ensure that the correlations between word-overlap metrics and human judgements were comparable across datasets, we standardized the processing of the evaluation dataset from Liu et al. (2016). In particular, the original data from Liu et al. (2016) has a token (either '<first_speaker>', '<second_speaker>', or '<third_speaker>') at the beginning of each utterance. This is an artifact left-over by the processing used as input to the hierarchical recurrent encoder-decoder (HRED) model (Serban et al., 2016a). Removing these tokens makes sense for establishing the ability of word-overlap models, as they are unrelated to the content of the tweets.

| Metric | Spearman | Pearson |
|--------|----------|---------|
| BLEU-1 | -0.026 (0.80) | 0.016 (0.87) |
| BLEU-2 | 0.065 (0.52) | 0.080 (0.43) |
| BLEU-3 | 0.139 (0.17) | 0.088 (0.39) |
| BLEU-4 | 0.139 (0.17) | 0.092 (0.36) |
| ROUGE | -0.083 (0.41) | -0.010 (0.92) |

Table 10: Correlations between word-overlap metrics and human judgements on the dataset from Liu et al. (2016), after removing the speaker tokens at the beginning of each utterance. The correlations are even worse than estimated in the original paper, and none are significant.

We perform this processing, and report the updated results for word-overlap metrics in Table 10. Surprisingly, almost all significant correlation disappears, particularly for all forms of the BLEU score. Thus, we can conclude that the word-overlap metrics were heavily relying on these tokens to form bigram matches between the model responses and reference responses.

**Evaluation speed** An important property of evaluation models is speed. We show the evaluation time on the test set for ADEM on both CPU and a Titan X GPU (using Theano, without cudNN) in Table 11. When run on GPU, ADEM is able to evaluate responses in a reasonable amount of time (approximately 2.5 minutes). This includes the time for encoding the contexts, model responses, and reference responses into vectors with the hierarchical RNN, in addition to computing the PCA projection, but does not include pre-training with VHRED. For comparison, if run on a test set of 10,000 responses, ADEM would take approximately 45 minutes.

| Metric | Wall time |
|--------|-----------|
| ADEM (CPU) | 2861s |
| ADEM (GPU) | 168s |

Table 11: Evaluation time on the test set.

This is significantly less time consuming than setting up human experiments at any scale. Note that we have not yet made any effort to optimize the speed of the ADEM model.

**Learning curves** To show that our learning procedure for ADEM really is necessary, and that the embeddings produced by VHRED are not sufficient to evaluate dialogue systems, we plot the Spearman and Pearson correlations on the test set as a function of the number of epochs in Figure 7. It is clear that, at the beginning of training, when the matrices $M$ and $N$ have been initialized to the identity, the model is incapable of accurately predicting human scores, and its correlation is approximately 0.

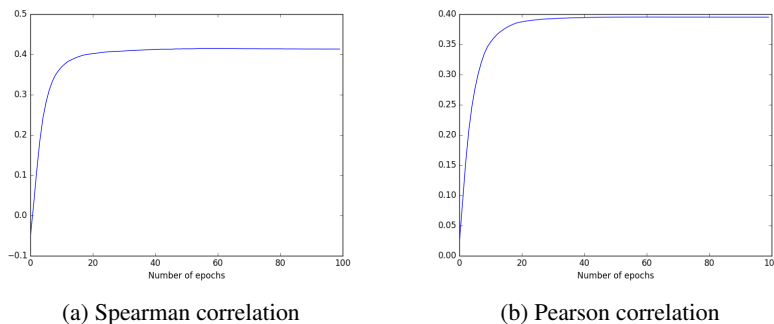

| (a) Spearman correlation | (b) Pearson correlation |

Figure 7: Plots showing the Spearman and Pearson correlations on the test set as ADEM trains. At the beginning of training, the model does not correlate with human judgements.

| Context | Reference response | Model response | Human score | \|BLEU-2\| score | \|ROUGE\| score | \|ADEM\| score |
|---|---|---|---|---|---|---|
| what theme do you guys want next on tumblr? we've had mariskamommymoments what do you want to see next? → im sorry. hope you feel better soon! → it will wear off. just hate feeling like this → im sure it will! just relax and take your time → i'm okay. just overly tired | maybe you need a bit more sleep | i'm really impressed. first guy to said that p what's time? sleep late its not good. i'm worried | 4 | 2.53 | 5.0 | 1.0 |
| 1 some pm syria - the editor of syrian government daily tishrin said she had been sacked over remarks to al <url> → replaced by business editor! sounds like the states lol | msm is very similar the world over ! tied to the government . | i'm not sure if i'm going to be able to see the <unk>. i'm not sure if i should be able to see it. | 4 | 2.53 | 4.75 | 1.22 |
| wonder how long it will take wayne rooney to get himself into trouble on twitter. odds? → probably. a long time. because i used the address in the paper to find the page and it doesn't exist! → here you go | thanks . i wonder why it didn't work ! | thanks let me see if this one works :p | 5 | 2.53 | 4.24 | 1.53 |

Table 12: Examples where a human and either BLEU-2 or ROUGE (after normalization) score the model response highly ($> 4/5$), while the ADEM model scored it poorly ($< 2/5$). These examples are drawn randomly (i.e. no cherry-picking). The bars around |metric| indicate that the metric scores have been normalized.

**Failure analysis** We now conduct a failure analysis of the ADEM model. In particular, we look at two different cases: responses where both humans and (normalized) ROUGE or BLEU-2 score highly (a score of 4 out of 5 or greater) while ADEM scores poorly (2 out of 5 or lower), and the converse, where ADEM scores the response highly while humans and either ROUGE or BLEU-2 score it poorly. We randomly sample (i.e. without cherry picking) three examples of each case, which are shown in Tables 12-13.

From Table 12, the cases where ADEM misses a good response, we can see that there are a variety of reasons for this cause of failure. In the first example, ADEM is not able to match the fact that the model response talks about sleep to the reference response or context. This is possibly because the utterance contains a significant amount of irrelevant information: indeed, the first two sentences are not related to either the context or reference response. In the second example, the model response does not seem particularly relevant to the context — despite this, the human scoring this example gave it $4/5$. This illustrates one drawback of human evaluations; they are quite subjective, and often have some noise. This makes it difficult to learn an effective ADEM model. Finally, ADEM is unable to score the third response highly, even though it is very closely related to the reference response.

We can observe from the first two examples in Table 13, where the ADEM model erroneously ranks the model responses highly, that ADEM is occasionally fooled into giving high scores for responses that are completely unrelated to the context. This may be because both of the utterances are short, and short utterances are ranked higher by humans in general since they are often more generic (as detailed in Section 5). In the third example, the response actually seems to be somewhat reasonable given the context; this may be an instance where the human evaluator provided a score that was too low.

| Context | Reference response | Model response | Human score | \|BLEU-2\| score | \|ROUGE\| score | \|ADEM\| score |
|---------|--------------------|----------------|-------------|------------------|-----------------|----------------|
| rage slightly dissipated. wouldn't have bothered restoring my phone but i need it to moan at tomorrow. → speaking of moaning. i'm actually going to email that chap that letter right now. → good plan | i 'm going to do a little wee blog about it too . all nice and measured , of course . | some. some unfortunately | 2 | 2.53 | 1.57 | 4.38 |
| high school flings college relationships → it seems like the other way around from wat i've seen | word . i 've seen a little of both . more of the college though | king james | 1 | 2.53 | 1.57 | 5.0 |
| is it getting light outside? i swear it looks blue. → time to go to sleepppp.. | for you , i 'm staying up | i'm going to the beach. | 1 | 2.53 | 1.57 | 5.0 |

Table 13: Examples where a human and either BLEU-2 or ROUGE (after normalization) score the model response low ($< 2/5$), while the ADEM model scored it highly ($> 4/5$). These examples are drawn randomly (i.e. no cherry-picking). The bars around |metric| indicate that the metric scores have been normalized.

**Data efficiency**     How much data is required to train ADEM? We conduct an experiment where we train ADEM on different amounts of training data, from 5% to 100%. The results are shown in Table 14. We can observe that ADEM is very data-efficient, and is capable of reaching a Spearman correlation of 0.4 using only half of the available training data (1000 labelled examples). ADEM correlates significantly with humans even when only trained on 5% of the original training data (100 labelled examples).

| Training data % | Spearman | p-value | Pearson | p-value |
|-----------------|----------|---------|---------|---------|
| 100 % of data | 0.414 | $< 0.001$ | 0.395 | $< 0.001$ |
| 75 % of data | 0.408 | $< 0.001$ | 0.393 | $< 0.001$ |
| 50 % of data | 0.400 | $< 0.001$ | 0.391 | $< 0.001$ |
| 25 % of data | 0.330 | $< 0.001$ | 0.331 | $< 0.001$ |
| 10 % of data | 0.245 | $< 0.001$ | 0.265 | $< 0.001$ |
| 5 % of data | 0.098 | 0.015 | 0.161 | $< 0.001$ |

Table 14: ADEM correlations when trained on different amounts of data.

