# Peer review of "Towards an automatic Turing test: Learning to evaluate dialogue responses"

_ICLR 2017 — rejected_

[Reviewer Comment · AnonReviewer3 · 01 Dec 2016]
**Please comment on more failure cases**

You give some qualitative results. But it would be really helpful to perform a more in-depth failure case analysis (e.g. give an overview of all important cases where ADEM is not well aligned with human judgement)

[Official Review · AnonReviewer3 · rating 7 · confidence 4 · 15 Dec 2016]
**The main idea of the paper is to learn the evaluation of dialogue responses in order to overcome limitations of current schemes such as BLEU**

Overall the paper address an important problem: how to evaluate more appropriately automatic dialogue responses given the fact that current practice to automatically evaluate (BLEU, METEOR, ...) is often insufficient and sometimes misleading. The proposed approach using an LSTM-based encoding of dialogue context, reference response and model response(s) that are then scored in a linearly transformed space. While the overall approach is simple it is also quite intuitiv and allows end-to-end training. As the authors rightly argue simplicity is a feature both for interpretation as well as for speed. 

The experimental section reports on quite a range of experiments that seem fine to me and aim to convince the reader about the applicability of the approach. As mentioned also by others more insights from the experiments would have been great. I mentioned an in-depth failure case analysis and I would also suggest to go beyond the current dataset to really show generalizability of the proposed approach. In my opinion the paper is somewhat weaker on that front that it should have been.

Overall I like the ideas put forward and the approach seems sensible though and the paper can thus be accepted.

[Official Review · AnonReviewer1 · rating 4 · confidence 4 · 19 Dec 2016]
**Why we should use yet another dialogue system as an evaluation metric?**

This paper propose a new evaluation metric for dialogue systems, and show it has a higher correlation with human annotation. I agree the MT based metrics like BLEU are too simple to capture enough semantic information, but the metric proposed in this paper seems to be too compliciated to explain.

On the other hand, we could also use equation 1 as a retrieval based dialogue system. So what is suggested in this paper is basically to train one dialogue model to evaluate another model. Then, the high-level question is why we should trust this model? This question is also relevant to the last item of my detail comments.

Detail comments:

- How to justify what is captured/evaluated by this metric? In terms of BLEU, we know it actually capture n-gram overlap. But for this model, I guess it is hard to say what is captured. If this is true, then it is also difficult to answer the question like: will the data dependence be a problem?
- why not build model incrementally? As shown in equation (1), this metric uses both context and reference to compute a score. Is it possible to show the score function using only reference? It will guarantee this metric use the same information source as BLEU or ROUGE. 
- Another question about equation (1), is it possible to design the metric to be a nonlinear function. Since from what I can tell, the comparison between BLEU (or ROUGE) and the new metric in Figure 3 is much like a comparison between the exponential scale and the linear scale.
- I found the two reasons in section 5.2 are not convincing if we put them together. Based on these two reasons, I would like to see the correlation with average score. A more reasonable way is to show the results both with and without averaging. 
- In table 6, it looks like the metric favors the short responses. If that is true, this metric basically does the opposite of BLEU, since BLEU will panelize short sentences. On the other hand, human annotators also tends to give short respones high scores, since long sentences will have a higher chance to contain some irrelevant words. Can we eliminate the length factor during the annotation? Otherwise, it is not surprise that the correlation.

[Official Review · AnonReviewer4 · rating 5 · confidence 4 · 22 Dec 2016]
**The idea is good, and the problem to address is very important. However, the proposed solution is short of desired one.**

This paper addresses the issue of how to evaluate automatic dialogue responses. This is an important issue because current practice to automatically evaluate (e.g. BLEU, based on N-gram overlap, etc.) is NOT correlated well with the desired quality (i.e. human annotation). The proposed approach is based on the use of an LSTM encoding of dialogue context, reference response and model response with appropriate scoring, with the essence of training one dialogue model to evaluate another model. However, the proposed solution depends on a reasonably good dialogue model to begin with, which is not guaranteed, rendering the new metric possibly meaningless.

[Author Response · Ryan Lowe · 23 Dec 2016]
**Comment to all reviewers**

We thank the reviewers for taking the time to comment. All of the reviewers agree that we are tackling a very important problem (dialogue response evaluation) that has been under-studied in the literature. We would like to emphasize to all reviewers that, while our paper is not perfect, we believe it is a strong first step towards addressing this problem. Further, we do not claim that our proposed evaluation metric should be the *only* automatic metric to be used, but rather we recommend it be *added* to existing evaluation metrics (since, unlike existing metrics, it actually correlates with human judgements). We are of the opinion that keeping the ‘good’ unpublished while waiting for the ‘perfect’ is not a productive way forward for science.

[Public Comment · (anonymous) · 24 Dec 2016 (modified: 26 Jan 2017)]
**Concern about Reviewer #4**

Dear ACs,

We are concerned about the quality of review provided by Reviewer #4. The only point of criticism offered by the reviewer is that "the proposed solution depends on a reasonably good dialogue model to begin with, which is not guaranteed, rendering the new metric possibly meaningless." We believe the review indicates that the reviewer did not thoroughly read or understand our paper, for several reasons:

1) As stated in the rebuttal, the claim that our model 'depends on a good dialogue model to begin with' is false. This can be seen by simply looking at our main results table, where we have additional results using publicly available sentence embedding methods (i.e. tweet2vec). What our method does require is a sentence embedding method, but we believe this is hardly a limitation of the work. 

2) Several details in the reviewer's summary of our paper are incorrect (it is not simply an LSTM used to encode the context, but a hierarchical model; our evaluation method is not in itself a dialogue model, as explained to Reviewer #1), although we do not mention this in our rebuttal. 

3) As stated in the rebuttal, we show that our evaluation procedure works (is correlated with human judgements) on a dataset (Twitter) that has seen the significant use in the recent literature. It is obvious that our method works on this popular dataset (since we demonstrate it in our experiments); thus, the claim that our method is 'meaningless' is not accurate (it would be very useful even if it only worked for Twitter).

4) Finally, the domain in which the reviewer implies our method would be 'meaningless' (when there exists no sentence embedding method for that domain) is very small: it encompasses only domains that (a) have too little data to train an embedding method (and if this is the case, it likely has too little data to train a dialogue model anyway), and (b) are different enough from all existing domains such that transfer learning can't be used (i.e. one can't use a generic embedding method such as skip-thought). The majority of domains studied in the literature (Twitter, Ubuntu, Reddit, movies, etc.) do not satisfy these criteria.

This is not to say there are no valid criticisms of our work -- indeed, the review provided by Reviewer #1 (although we believe they are mistaken) at least indicates that the reviewer read the paper. 

We would appreciate if this could be taken into consideration when deciding whether to accept or reject this paper.


Thanks,

-The Authors

[Author Response · Ryan Lowe · 14 Jan 2017]
**Updated paper accounting for reviewer comments**

We thank the reviewers again for their feedback. While our detailed rebuttal to each reviewer is written as as a direct response to the review, we have now incorporated many of the reviewers' comments into an updated version of the paper. In particular, we have addressed the following items:

1) We have made several alterations to alleviate possible misunderstandings from reading the paper. Most notably, we have added a paragraph to Section 4 explaining why the ADEM model is not a dialogue retrieval model. 

2) As recommended by Reviewer #1, we have produced two additional results that help clarify the workings of different parts of the model. In Table 3, we show correlations for the ADEM model when it can only compare the model response to the context (C-ADEM), and when it can only compare the model response to the reference response (R-ADEM). It seems that ADEM is mostly using the comparison to the reference response in order to assign its score. This makes sense, since the reference response is often closer semantically to the model response than the context, and goes to further illustrate how our model is different from a dialogue retrieval model (which could only use the context, i.e. C-ADEM). 

3) We provide new correlation results for word overlap metrics on the dataset from Liu et al. (2016). In particular, we standardized the pre-processing procedure by removing the

[Final Decision · Program Chairs · 06 Feb 2017]
**ICLR committee final decision**

Noting the authors' concern about one of the reviewers, I read the paper myself and offer my own brief review.
 
 Evaluation is an extremely important question that does not get enough attention in the machine learning community, so the authors' effort is welcomed. The task that the authors are trying to evaluate is especially hard; in fact, it is not even clear to me how humans make these judgments. The low kappa scores on some of the non-"overall" dimensions, and only moderate agreement on "overall," are quite worrisome. What makes a good "chat" dialogue? The authors seem not to have qualitatively grappled with this key question, rather defining it empirically as "whatever our human judges think it is." This is, I think the deepest flaw of the work; the authors are rushing to automate evaluation without taking the time to ponder what good performance actually is. 
 
 That aside, I think the idea of automatic evaluation as a modeling problem is worth studying. The authors note that this has been done for other problems, such as machine translation. They give only a cursory discussion of this prior work, however, and miss the seminal reference, "Regression for sentence-level MT evaluation with pseudo references," Albrecht, Joshua, and Rebecca Hwa, ACL 2007.
 
 The paper would be much stronger with some robustness analysis; does the quality of the evaluation hold up if the design decisions are made differently, if less data are used to estimate the evaluation model, etc.? How does it hold up across datasets, and across different types of dialogue systems? As a methodological note, there are a lot of significance tests here and no mention of any attempt to correct for this (e.g., Bonferroni, FDR, etc.).
 
 As interesting as the ideas here are, I can't see the dialogue community rushing to adopt this approach to evaluation based on the findings in this paper. I do think that the ideas it presents should be hashed out in a public forum sooner rather than later, and therefore recommend it as one of a few papers to be presented at the workshop.